# Vaccination to prevent T cell subversion can protect against persistent hepacivirus infection

Alex S. Hartlage[1,2], Satyapramod Murthy[1], Arvind Kumar[1], Sheetal Trivedi[1], Piyush Dravid[1], Himanshu Sharma[1], Christopher M. Walker[1,3] & Amit Kapoor[1,3]

Efforts to develop an effective vaccine against the hepatitis C virus (HCV; human hepacivirus) have been stymied by a lack of small animal models. Here, we describe an experimental rat model of chronic HCV-related hepacivirus infection and its response to T cell immunization. Immune-competent rats challenged with a rodent hepacivirus (RHV) develop chronic viremia characterized by expansion of non-functional CD8+ T cells. Single-dose vaccination with a recombinant adenovirus vector expressing hepacivirus non-structural proteins induces effective immunity in majority of rats. Resolution of infection coincides with a vigorous recall of intrahepatic cellular responses. Host selection of viral CD8 escape variants can subvert vaccine-conferred immunity. Transient depletion of CD8+ cells from vaccinated rats prolongs infection, while CD4+ cell depletion results in chronic viremia. These results provide direct evidence that co-operation between CD4+ and CD8+ T cells is important for hepacivirus immunity, and that subversion of responses can be prevented by prophylactic vaccination.

[1] Center for Vaccines and Immunity, The Research Institute at Nationwide Children's Hospital, Columbus, OH 43205, USA. [2] Medical Scientist Training Program, College of Medicine and Public Health, The Ohio State University, Columbus, OH 43210, USA. [3] Department of Pediatrics, College of Medicine and Public Health, The Ohio State University, Columbus, OH 43210, USA. Correspondence and requests for materials should be addressed to A.K. (email: amit.kapoor@nationwidechildrens.org)

Chronic liver infections caused by the hepatitis C virus, a blood-borne human hepacivirus, are a major cause of serious progressive liver diseases and affect 71 million people worldwide[1]. Although curative antivirals are now available, these agents are unaffordable to most, ineffective in those with advanced liver disease, and do not protect against viral reinfections[2,3]. Development of a vaccine to prevent HCV chronicity in naïve or cured individuals could substantially reduce viral transmission and disease rates, and thus remains a major unmet clinical need.

Acute HCV infections spontaneously resolve in 20–30% of cases[4]. Although mechanisms of protective immunity remain uncertain, a substantial body of evidence implicates a critical role for T cells in this process. Indeed, termination of HCV viremia in humans and chimpanzees is kinetically associated with the onset of sustained virus-specific CD4+ and CD8+ T cell responses, which are notably absent or dysfunctional in those who fail to control virus[5–11]. CD8+ T cells with limited effector functions are often present during persistent infections but are exhausted or target regions of the virus that have developed escape mutations in MHC class I epitopes[12–17]. Likelihood of spontaneous HCV clearance is also strongly correlated with MHC class I and II molecule diversity[18–21]. In addition, successful resolution of primary HCV results in the formation of long-lived memory T cells that are rapidly recalled during reinfection; in chimpanzees, these responses were demonstrated to be necessary for control of tertiary infection[22–24]. Finally, a T cell vaccine that primed HCV-specific CD4+ and CD8+ T cell responses in chimpanzees led to suppressed viremia and accelerated virus control after experimental challenge[25]. Despite these circumstantial lines of evidence, however, a direct causal relationship between a host T cell response and primary HCV infection outcome has yet to be established. Moreover, whether T cells elicited via vaccine are sufficient to prevent HCV persistence in humans in the absence of neutralizing antibodies is unclear, although testing of this concept is underway (ClinicalTrials.gov NCT01436357).

Animal models for the testing of HCV immunity and vaccination concepts are limited. Chimpanzees, the only species besides humans fully susceptible to HCV, have been phased out from medical research. Furthermore, mice that have been experimentally manipulated to support HCV infection through expression of human entry factors or engraftment with human hepatocytes require ablation of innate or adaptive immunity for virus to persist[26].

Recently, we described an HCV-related rodent hepacivirus (RHV) that infects immune-competent laboratory mice and rats[27,28]. Discovered in feral rats in New York City[29], RHV is a close relative of HCV and displays key similarities in genomic organization, RNA secondary structure, and polyprotein processing. RHV also possesses two liver-specific microRNA-122 binding sites in the 5′ UTR that are a defining characteristic of the HCV genome. In mice, RHV is cleared rapidly within weeks, making them unsuitable for studies of HCV persistence and vaccine testing. In contrast, RHV spontaneously persists at high frequency in rats, the natural host of the virus. This occurs despite the activation of hepatic innate and adaptive immune processes similarly to chronic HCV in humans[28,30]. Protective mechanisms that fail to control RHV in rats and whether they can be successfully elicited via immunization has yet to be investigated.

Here, we provide the first direct evidence linking an inadequate T cell response to the persistence of a primary hepacivirus infection in a natural host species. Using rats infected with RHV as a surrogate model of chronic HCV infection in humans, we demonstrate that (a) spontaneous hepacivirus persistence is associated with a transient virus-specific CD8+ T cell response that develops in the absence of strong CD4+ T cell help, (b) pre-exposure priming of virus-specific T cells by an adenovirus vector expressing non-structural viral proteins prevents virus persistence in majority of animals, and (c) antibody-mediated depletion of vaccine-primed T cells abrogates protective immunity after challenge. Thus, immunological strategies designed to invigorate or protect virus-specific T cells from immune failure during infection may help reduce global HCV transmission and disease.

## Results

**T cell immunity during hepacivirus persistence.** In contrast to mice, challenge of laboratory rats with RHV produces a hepatotropic infection that spontaneously persists irrespective of strain background[28]. In order to define cellular immunity to RHV in its natural host setting, we infected inbred Lewis rats, which are frequently utilized for studies of infectious disease and possess a well-defined histocompatibility complex[31,32]. Intravenous challenge of rats with RHV-containing inoculum ($10^6$ genomes) caused a high-titer chronic infection capable of persisting through 50 weeks (Fig. 1a). Viremia and serum ALT levels, a marker of acute liver damage, were largely stable after infection, with only a small drop in serum viral titers (~1–2 log) occurring 6–10 weeks after challenge. This partial suppression of viremia was temporally associated with the onset of IFNγ ELISpot responses targeting the RHV core protein in both liver and spleen (Fig. 1b). Intracellular cytokine staining for IFNγ revealed this functional response to be mediated solely by CD4+ T cells (Fig. 1c). An example of our gating strategy for identification of cytokine positive cells can be seen in Supplementary Figure 1. Notably, core-specific CD4+ T cells producing IFNγ were present at substantially higher frequency when compared to responses directed against the RHV non-structural proteins (NS3-4B), which were largely absent (Fig. 1c). A subset of core-specific cells was multifunctional as evidenced by production of both IFNγ and TNFα (Fig. 1d) and secretion of IL-2 (Fig. 1e), although most (>75%) were single cytokine producers (Fig. 1f). In addition, this core-specific helper response exhibited remarkable breadth, targeting at least seven MHC class II epitopes located in the carboxyl half of the protein (Fig. 1g, h; Supplementary Table 1).

Functional detection of RHV-specific CD8+ T cell responses was not achieved at any time-point after challenge (Fig. 1c, d) nor could short-term lymphocyte lines be successfully driven from recovered liver tissue. In addition, potential escape mutations were not observed in defined MHC class I epitopes (identification of these are described in detail below), providing further evidence that CD8+ T cells lack sufficient effector function to exert selective pressure against persistent virus. To determine whether CD8+ T cells were primed by replicating virus yet non-functional, we tracked virus-specific CD8+ T cells using an MHC class I tetramer containing the RT1-A$^l$-restricted NS3$_{1497}$ epitope. Tetramer responses were first visualized in the infected liver 14 days after infection yet contracted sharply below detection by day 21 (Fig. 2a). As expected, these responses were heavily dysfunctional, producing no cytokine after stimulation with NS3$_{1497}$ peptide (Fig. 2b). The rapid disappearance of the virus-specific population combined with its absence of effector activity prompted us to examine expression of markers for activation and apoptosis (Supplementary Fig. 2). Granzyme B, a cytolytic enzyme upregulated in effector CD8+ T cells, was expressed by ~40% of NS3$_{1497}$-specific cells at day 14 (Fig. 2c, d), indicating the possibility for cytotoxic activity although none could be detected by in vivo killing assay (Supplementary Fig. 3). Upregulation of the adhesion molecule CD44H, a general indicator of cellular activation, was also detected in majority (~70%) of the NS3$_{1497}$-specific population (Fig. 2c, d). Surprisingly, little to none of the

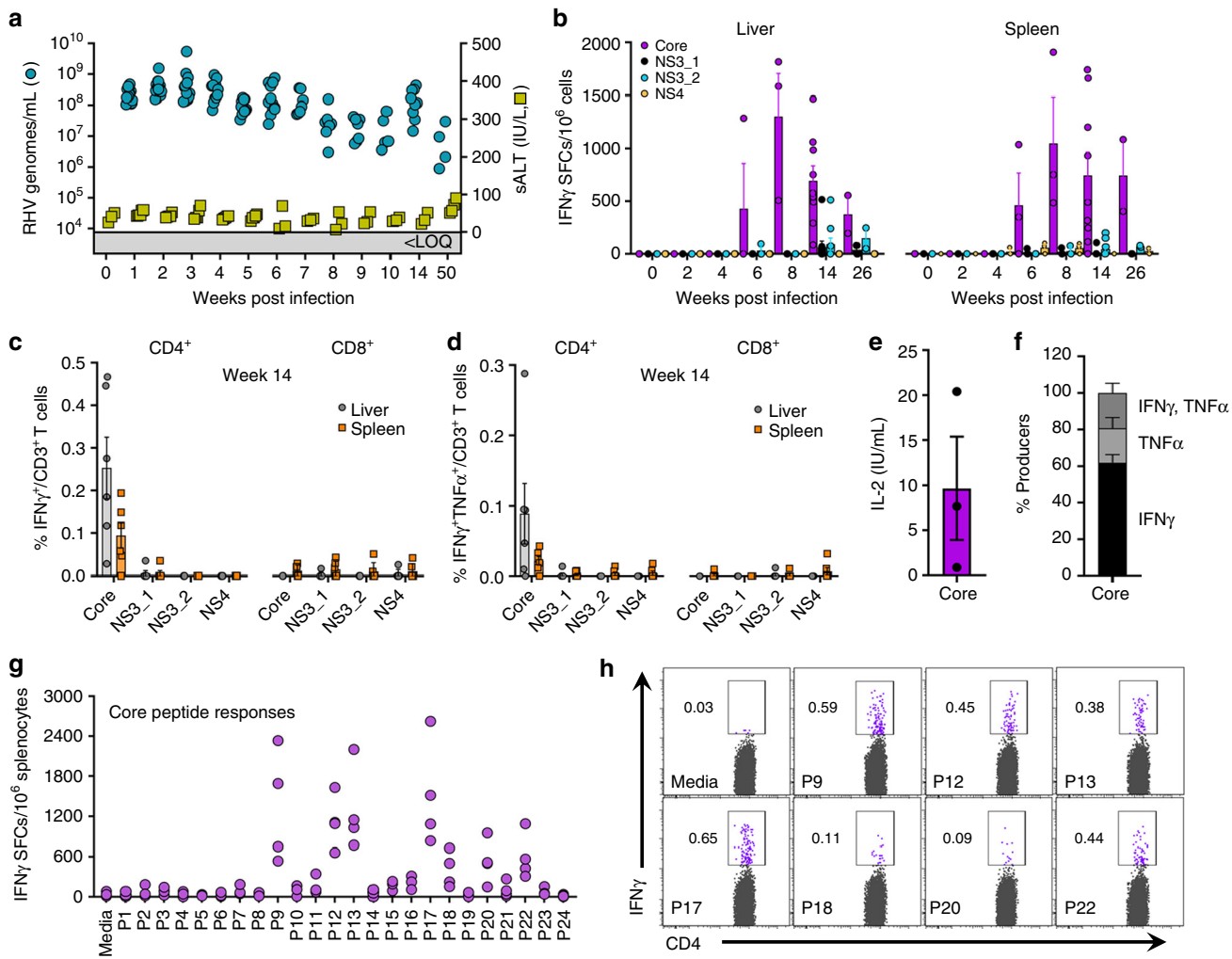

**Fig. 1** RHV infection is associated with a limited virus-specific T cell response. Lewis rats were challenged intravenously with $10^6$ genomes of RHV and analyzed for infection, pathology, and cellular immunity. **a** Viremia (blue circles) and serum ALT (sALT, yellow squares) levels after challenge. Solid line with shading below indicates limit of quantification (LOQ) of RT-PCR. **b** IFNγ ELISpot responses in livers (left panel) and spleens (right panel) of rats at various time points post infection. Mononuclear cells were stimulated with viral peptides (2 μg/mL) for 40–48 h. Data from $n = 2$–9 rats per time point are shown (mean±SEM). **c, d** Frequencies of IFNγ⁺ and IFNγ⁺TNFα⁺ T cells in livers (gray circles) and spleens (orange squares) after 5-h peptide stimulation (2 μg/mL). Rats were infected for 14 weeks prior to analysis. Left and right panels show frequencies in CD4⁺ and CD8⁺ T cells, respectively. Data from $n = 6$ rats are shown (mean±SEM). **e** Core peptide-specific IL-2 secretion by CD4⁺ splenocytes. Cells were stimulated with core peptides (2 μg/mL) for 48-h prior to supernatant collection. Rats were infected for 14 weeks prior to analysis. Responses from $n = 3$ rats are shown (mean±SEM). **f** Relative frequency of intrahepatic CD4⁺ T cells producing IFNγ, TNFα, or both after 5-h stimulation with core peptides (2 μg/mL). Data from $n = 6$ rats are shown (mean±SEM). **g** Splenic IFNγ ELISpot responses against 24 individual core peptides (10 μg/mL). Responses from $n = 4$ rats at >6 weeks infection are shown. **h** Flow cytometric analysis of IFNγ production by CD4⁺ T cells following 5-h stimulation with core-derived MHC class II epitopes (10 μg/mL)

NS3₁₄₉₇-specific population was apoptotic as measured by annexin V staining (Fig. 2c, d), suggesting an alternative mechanism for their premature contraction. Notably, expression of common inhibitory molecules such as PD-1 could not be assessed in this setting due to the unavailability of cross-reactive antibodies for rats. Taken together, these observations demonstrate that acute RHV infection is associated with an expansion of non-functional virus-specific CD8⁺ T cells that collapse to undetectable levels as persistence spontaneously develops. Notably, this secondary failure of the CD8⁺ T cell response occurs before the late appearance of cytokine-producing CD4⁺ T cells specific for core.

**Recombinant adenovirus vaccine primes RHV-specific T cells.** Failure to generate or sustain functional CD4⁺ and CD8⁺ T cell

responses, most notably against HCV non-structural proteins that are dominant targets of cellular immunity, is a reliable predictor of HCV persistence in humans[11]. We therefore hypothesized that a similar deficit in T cell immunity against RHV non-structural proteins may contribute to virus persistence. To test this, we constructed a recombinant human adenovirus serotype 5 (Ad5) vector that expresses the RHV NS3-5B proteins (Ad-NSmut), with the objective of priming protective T cell responses. Importantly, this approach avoids generation of neutralizing antibody responses against the RHV E1 and E2 envelope proteins, which may influence RHV replication and persistence. Of note, a similar immunization strategy elicited robust T cell immunity against HCV non-structural proteins in chimpanzees[25] and humans[33].

In Lewis rats, strong T cell responses against the RHV non-structural proteins were observed in the spleen 14 days after

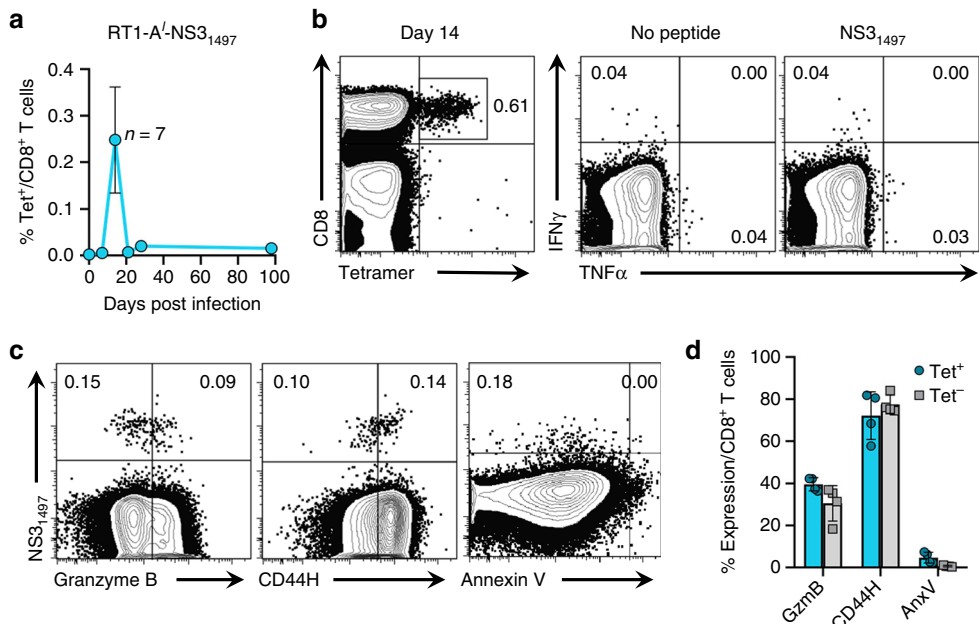

**Fig. 2** Expansion of non-functional RHV-specific CD8+ T cells during infection. Lewis rats were challenged intravenously with $10^6$ genomes of RHV and intrahepatic CD8+ T cell responses to the RT1-A$^l$-restricted NS3$_{1497}$ epitope were assessed by class I tetramer and cytokine production assays.
**a** Frequency of intrahepatic CD8+ T cells specific to NS3$_{1497}$ epitope at the indicated time post infection. Data from $n = 3$–7 rats per timepoint are shown (mean±SEM). **b** Sample flow plots showing frequency of intrahepatic NS3$_{1497}$-specific CD8+ T cells at day 14 when analyzed by tetramer staining (left) versus production of intracellular cytokines (right). For stimulation, cells were pulsed for 5-h with minimal peptide (10 μg/mL). **c** Representative flow plots of intrahepatic CD8+ T cells stained with NS3$_{1497}$ tetramer and antibodies against granzyme B, CD44H, and annexin V at day 14. **d** Calculated frequencies of granzyme B (GzmB), CD44H, and Annexin V (AnxV) expressing cells within the tetramer (Tet) positive (blue circles) and negative (gray squares) CD8+ T cell populations 14 days post infection. Data from $n = 4$ rats are shown (mean ± SEM). Numbers within all FACS plots indicate the frequency of positive cells within the CD8+ T cell gate

a single intramuscular dose of Ad-NSmut vector ($5 \times 10^8$ ifu; Supplementary Fig. 4a). The vector-elicited response was comprised of CD4+ and CD8+ T cells targeting at least eleven MHC class II epitopes and two class I epitopes located across the NS3-5B proteins, respectively (Supplementary Fig. 4b–e). Majority of these targeted epitopes are located in the carboxyl terminus of the NS3 protein. One of these epitopes (NS3$_{1497}$) was successfully incorporated into a functional MHC class I tetramer as described above. To increase detection of functional responses after immunization, these epitopes were combined together at high concentration into a single testing pool. Indeed, a high frequency of RHV-specific CD8+ T cells targeting dominant epitopes was detected in liver after 14 days (Fig. 3a). A CD4+ T cell response targeting class II epitopes was also detected in liver, but at relatively low frequencies (Fig. 3a). Together, these data demonstrate the capacity of a vaccine vector to elicit virus-specific T cell immunity that is qualitatively different than that primed by persistent RHV.

**T cell immunization prevents hepacivirus persistence**. To assess the in vivo protection conferred by Ad-NSmut, rats were challenged intravenously with RHV 21 days after immunization. Compared to empty control vector (Ad-null), Ad-NSmut immunization successfully countered development of persistent viremia in 6 out of 9 challenged rats (Fig. 3c, d); sterilizing immunity was not observed in any animal. The statistical significance for 0/6 versus 6/9 using Fisher's exact test is $p = 0.0278$. In infection resolvers, termination of viremia occurred within 14 days of virus challenge and was associated with small, transient rises in serum ALT levels in a subset of animals (Supplementary Fig. 5). Rats 560 and 561, which failed to resolve infection, exhibited no substantial control of virus after challenge. Potential

MHC class I escape mutations, which could contribute to inadequate virus control, were not detected in these two rats. Rat 558 followed a more unique course. A short duration of strong antiviral control characterized by an oscillating pattern of viremia was observed before stable, high-titer persistence was ultimately established after 21 days (Fig. 3d). Interestingly, rebound of viremia was temporally linked to the emergence of a non-synonymous mutation in the NS4A$_{1578}$ MHC class I epitope described above (Fig. 3e). This methionine to valine mutation at amino acid position 4 substantially reduced peptide recognition by virus-specific CD8+ T cells ex vivo (Fig. 3f), suggesting virus escape driven by host selection pressure. Mutation of the NS3$_{1497}$ class I epitope was not observed. An additional non-synonymous mutation was also detected in an 18 amino acid-long sequence located near the amino terminus of NS3. This region was finely mapped and found to contain an unknown CD8+ T cell epitope (NS3$_{974}$) that was not predicted based upon class I binding potential (Fig. 3e, Supplementary Fig. 6). As previously noted, mutation of these epitopes was not observed during infection of naïve rats, indicating that selection of these MHC class I escape variants was the direct result of an Ad-NSmut-driven CD8+ T cell response.

**Functional recall of T cell responses after challenge**. We next assessed the functional recall of CD4+ and CD8+ T cells during vaccine-mediated resolution of RHV. In infection resolvers, vector-mediated control was characterized by an ~2-fold expansion in the frequency of CD8+ T cells producing IFNγ in liver at day 9 post infection compared to unchallenged rats (Figs. 4a and 3c). The proportion of CD8+ T cells producing IFNγ or TNFα remained relatively unaltered during this expansion, with ~40% showing multifunction (Fig. 4e). Consistent with our previous

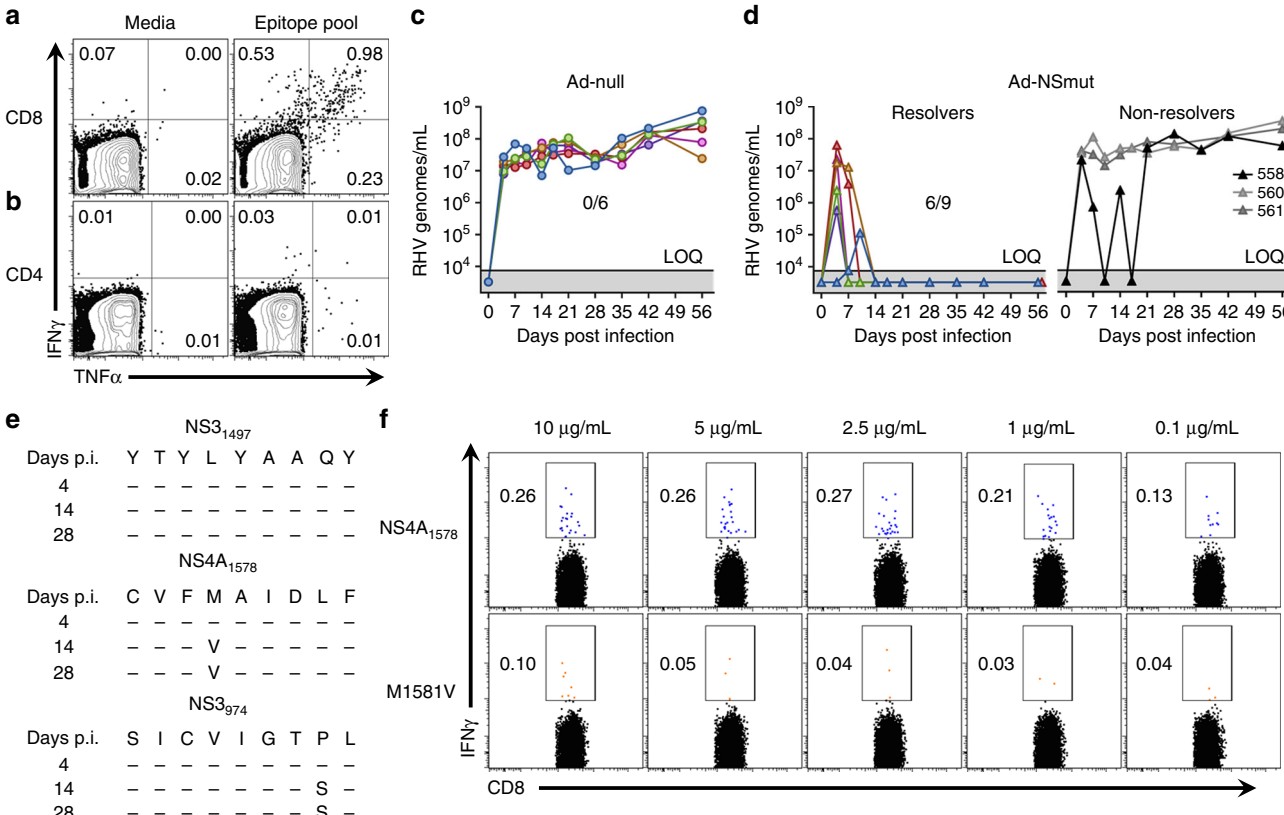

**Fig. 3** T cell immunization confers partial protection against RHV infection. Lewis rats were injected intramuscularly with $5 \times 10^8$ ifu Ad-NSmut or empty control vector (Ad-null). After 3 weeks, rats were challenged intravenously with $10^6$ genomes RHV and analyzed for infection and viral evolution. **a**, **b** Representative flow plots of intrahepatic cytokine production in CD8+ **a** and CD4+ **b** T cells 2 weeks after vaccination. Cells were stimulated for 5-h with a pool of MHC class I and II epitopes (5 μg/mL). Numbers shown indicate frequency of cytokine positive cells. **c**, **d** Replication kinetics and infection outcome in **c** Ad-null or **d** Ad-NSmut immunized rats. Solid line with shading below indicates limit of quantification (LOQ) of RT-PCR. **e** Evolution of MHC class I epitopes in rat 558. The NS3₉₇₄ epitope was putatively identified by sequencing analysis and subsequently confirmed/mapped by cytokine response assay. **f** Flow cytometric analysis of intracellular IFNγ production by CD8+ T cells in immune splenocytes after stimulation with intact NS4₁₅₇₈ epitope (top row) or peptide containing the M1581V mutation (bottom row). Cells were stimulated for 5-h with decreasing amounts of peptide as shown. Numbers shown indicate frequency of IFNγ+ cells within CD8+ T cell gate

findings, rats immunized with Ad-null vector failed to generate functional CD8+ T cell responses against identical NS3-5B epitopes during early (day 14) and late (day 100) phases of RHV infection (Supplementary Fig. 7a, c).

By comparison, the change in CD4+ T cell responses during resolution was far more robust. Total frequency of CD4+ T cells producing IFNγ increased >200-fold in the liver after challenge (Fig. 4b, d). Nearly all (>95%) of these responses were sequestered in liver when compared to spleen, a small subset of which (~25%) exhibited multi-cytokine function (Fig. 4f); lone producers of TNFα were not observed. CD4+ T cell responses against identical epitopes were generated in infected rats that had been immunized with Ad-null vector, although at substantially reduced frequencies and without multi-cytokine function (Supplementary Fig. 7b, d).

**CD4+ and CD8+ cells are required for vaccine efficacy.** To assess the relative importance of vaccine-elicited CD4+ versus CD8+ T cells in prevention of RHV persistence, immunized rats were treated with depleting antibodies targeting these subsets immediately prior to RHV challenge. Treatment of rats with anti-CD8α antibody (−2, 5, and 12 days post infection) resulted in >99.9% depletion of blood CD8+ T cells at time of challenge compared to IgG1 controls (Fig. 5a). As expected, mock-depleted rats were completely protected from RHV persistence (Fig. 5b). In

CD8α-depleted rats, however, virus control was delayed until blood CD8+ T cells began to recover at 28 days post infection (Fig. 5a, c). Interestingly, an oscillatory pattern of control was observed in these rats until virus was finally undetectable at 84 days post infection (Fig. 5c). Notably, absence of CD8+ cells did not fully ablate early CD4+ T helper cell responses in the immunized liver (Supplementary Fig. 8). These observations are consistent with CD8+ T cells being primary mediators of hepacivirus clearance[24].

Depletion of CD4+ T cells, although less efficient, produced a more profound effect. Treatment of rats with anti-CD4 antibody (−3, −1, and 4 days post infection) resulted in >95% depletion of blood CD4+ T cells at time of challenge compared to IgG2α controls (Fig. 4d). This early loss of CD4+ cells resulted in persistent infection in all rats even after near complete recovery of the blood CD4+ T cell compartment (Fig. 5d–f). A partial suppression of virus was initially present after infection, but was subsequently lost at 17 days post infection. Notably, virus escape from CD8+ T cells was not a major factor driving immune failure in this setting since only a single CD4-depleted rat showed a potential MHC class I escape mutation in circulating virus after rebound of viremia (Supplementary Fig. 9). This finding further supports a central role for CD4+ T cell help in protection against persistent hepacivirus infections[7,8,23].

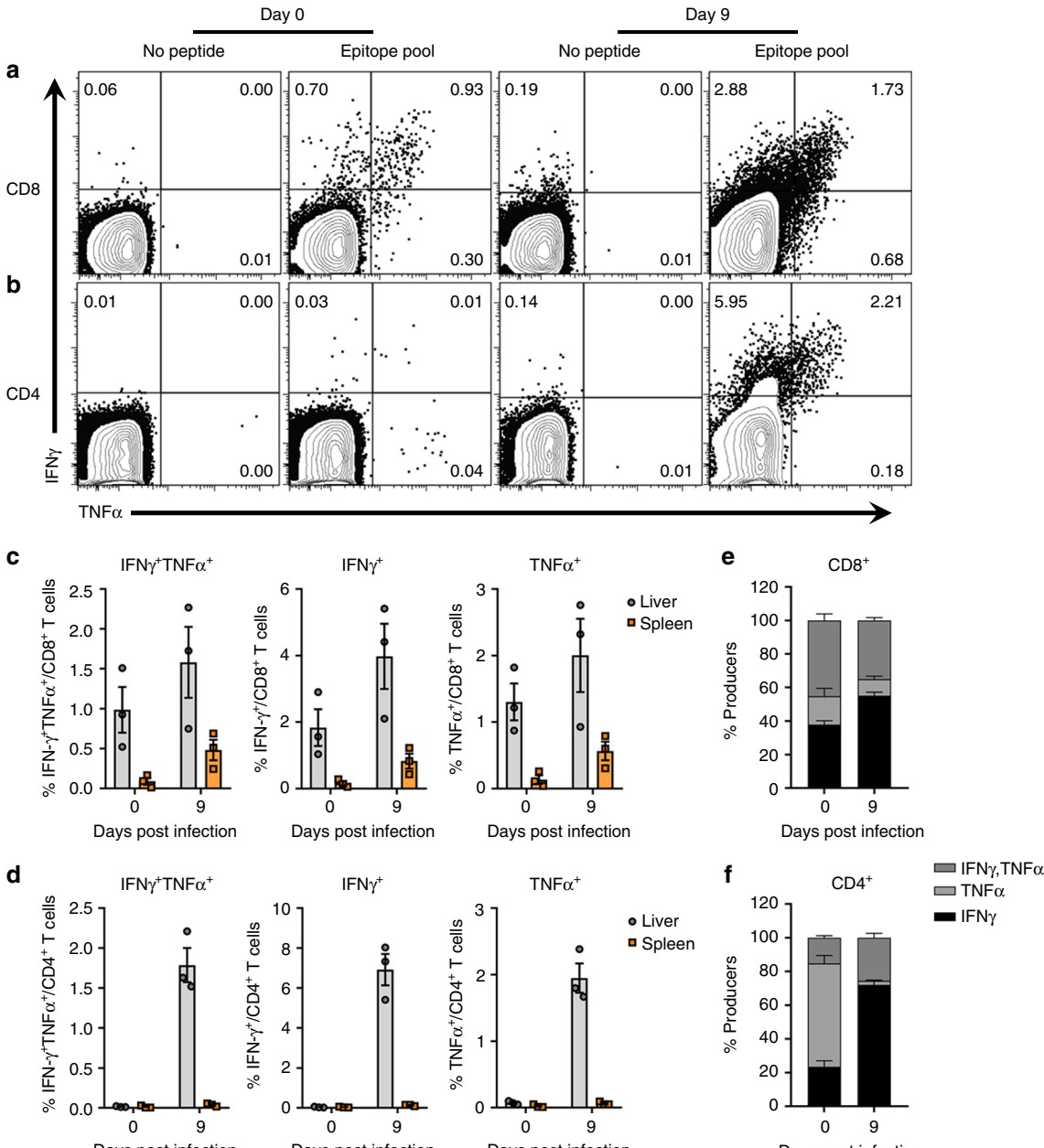

**Fig. 4** RHV-specific T cell responses during vaccine-mediated resolution. Lewis rats were immunized intramuscularly with 5 ×10$^8$ ifu Ad-NSmut. After 3 weeks, rats were challenged intravenously with 10$^6$ genomes RHV. At days 0 and 9 post infection, virus-specific T cell responses were quantified by flow cytometric analysis. **a**, **b** Representative flow plots of intrahepatic IFNγ and TNFα production in CD8$^+$ **a** and CD4$^+$ **b** T cells. Mononuclear cells were stimulated for 5-h with media alone or a pool of MHC class I and II epitopes (5 μg/mL). Numbers indicate frequency of cytokine positive cells within T cell gate. **c**, **d** Calculated frequencies of total and dual positive cytokine producers in CD8$^+$ **c** and CD4$^+$ **d** T cell compartments. Responses in liver (gray circles) and spleen (orange squares) are shown for comparison. Data from $n = 3$ rats per group are shown (mean±SEM). **e**, **f** Relative frequency of intrahepatic RHV-specific CD8$^+$ **e** or CD4$^+$ **f** T cells producing IFNγ or TNFα at 0 and 9 days post infection. Data from $n = 3$ rats per group are shown (mean±SEM)

## Discussion

We recently established a surrogate rat model of chronic HCV infection using an evolutionarily related hepacivirus identified in feral rats[28]. This study was undertaken to assess cellular immunity and to define protective mechanisms that spontaneously fail in the setting of hepacivirus persistence. Our results demonstrate that antiviral T cell immunity is a critical rheostat that determines hepacivirus infection outcome in a natural host species. Specifically, we show that a vaccine designed to prime protective T cell immunity without concomitant antibody responses to viral envelope proteins promoted the resolution of an otherwise persistent infection.

Long-term control of primary HCV infections is thought to be critically dependent upon the generation and maintenance of effective T cell immunity[11,34]. Indeed, the only HCV vaccine currently undergoing phase I/II testing in humans is a prime-boost approach using recombinant viral vectors encoding HCV non-structural proteins (ClinicalTrials.gov: NCT01436357). This strategy is unique for the vaccine field as it will determine whether memory T cells alone can prevent chronic infection by a

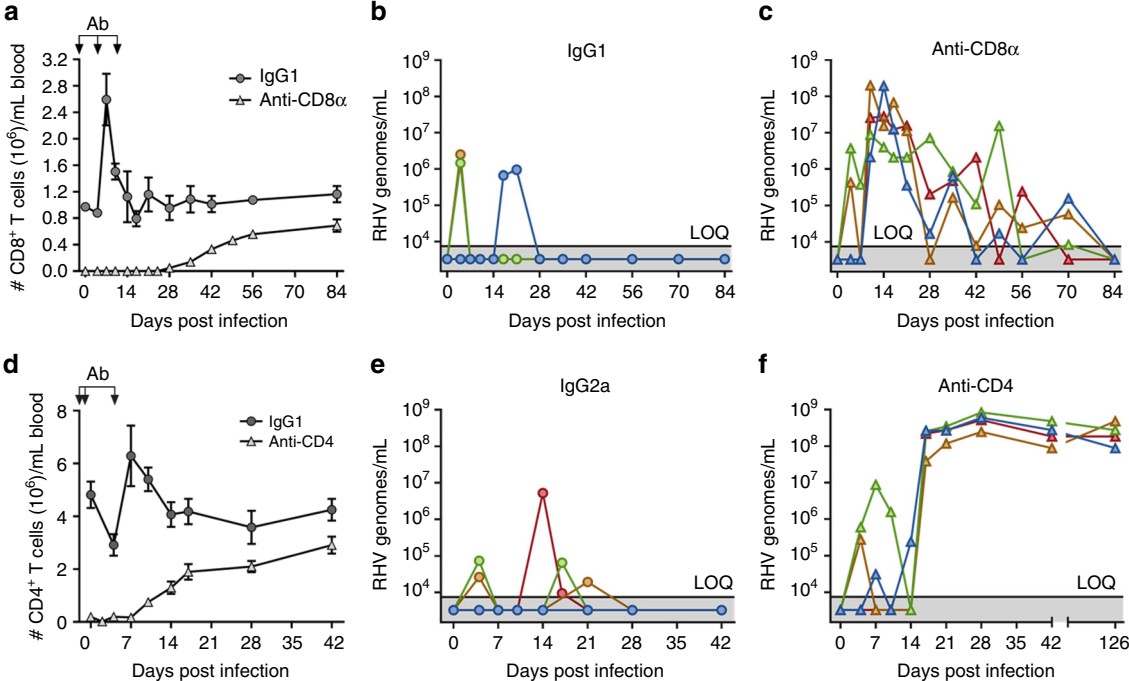

**Fig. 5** Vaccine-driven virus resolution is CD4+ and CD8+ cell-dependent. Lewis rats were injected intramuscularly with $5 \times 10^8$ ifu Ad-NSmut. After 3 weeks, rats were challenged intravenously with $10^6$ genomes RHV. **a–c** At days −2, 5, and 12 post infection, rats were transiently depleted of CD8α+ cells by antibody or isotype control. **a** Number of CD8+ T cells/mL blood of IgG1 and anti-CD8α antibody-treated rats as determined by flow cytometric analysis. Arrows indicate timing of antibody administration. Data from $n = 4$ rats per group are shown (mean ± SEM). **b, c** Viremia in IgG1 and anti-CD8α antibody treated rats. Solid line with below shading indicates limit of quantification (LOQ) of RT-PCR. **d–f** At days −3, −1, and 4 post infection, rats were transiently depleted of CD4+ cells by antibody or isotype control. **d** Number of CD4+ T cells/mL blood of IgG2a and anti-CD4 antibody-treated rats as determined by flow cytometric analysis. Arrows indicate timing of antibody administration. Data from $n = 4$ rats per group are shown (mean ± SEM). **e, f** Viremia in IgG2a and anti-CD4 antibody treated rats. Solid line with below shading indicates LOQ of RT-PCR

human pathogenic virus. Testing of this unconventional approach is supported by studies in chimpanzees where a similar vaccination strategy led to suppressed viral loads and accelerated HCV clearance following a heterologous virus challenge. However, whether T cell vaccination decreased the likelihood of HCV persistence could not be concluded from this study due to insufficient animal numbers. Our data are the first to conclusively demonstrate that a T cell vaccine can successfully alter the outcome of a primary hepacivirus infection that persists. Indeed, immunization with Ad-NSmut protected 6 out of 9 rats from RHV persistence compared to 0 out of 6 mock-vaccinated controls for a significance value of $p = 0.0278$. This represents not only an important proof-of-principle for human vaccination studies, but on more fundamental level, provides direct support to the concept that T cell failure is a cause rather consequence of hepacivirus persistence.

These data also provide insight into potential mechanisms of T cell failure following vaccination. Rat 558 transiently controlled RHV viremia after challenge but ultimately developed persistent infection. This was kinetically linked to the selection of non-synonymous mutations in two MHC class I epitopes. We showed that at least one of these mutated epitopes effectively extinguished CD8+ T cell recognition ex vivo, indicating escape from ongoing T cell pressure. Of note, mutations in these epitopes did not occur in unvaccinated rats during infection. These data suggest that although vaccination can induce a T cell response capable of controlling RHV infection, these cells may also create greater immune pressure for selection of escape variants[35]. Factors that may contribute to this outcome are insufficient help and narrow epitope breadth[23]. Importantly, protection from RHV persistence was assessed after only a single dose of Ad-NSmut. Thus, it is

possible that additional immunizations are necessary to boost responses to a threshold that are less susceptible to failure and provide more consistent protection[33,36]. This may be especially important in the cases of vaccinated rats 560 and 561, who showed no suppression of RHV viremia during infection.

A major challenge in the development of effective T cell-inducing vaccines against chronic viral infections such as HCV is an incomplete understanding of the features necessary for vaccine efficacy. Studies of T cell immunity from humans and animal models suggest a number of key parameters may be required. These include induction of both T cell subsets, targeting of multiple antigens, cytokine polyfunctionality, tissue localization, and rapid and powerful recall capability[37]. Ad-NSmut fulfilled each of these criteria although with some potential caveats. We found that single immunization with Ad-NSmut primed both CD4+ and CD8+ T cell responses specific to multiple epitopes spanning the NS3-5B protein. These responses homed to the liver and, in the case of CD8+ T cells, were heavily enriched compared to spleen. Due to the unavailability of various rat-specific antibodies, a more detailed assessment of the phenotype and function of these cells could not be performed. Nevertheless, cytokine production of IFNγ and TNFα was sufficiently analyzed. For CD8+ T cells, >40% of induced cells produced both cytokines, indicating strong polyfunctionality. In contrast, >80% of the CD4+ T cell response produced only a single cytokine, predominantly TNFα. This finding might indicate poor helper cell polyfunctionality, although we cannot discount the possibility that other critical cytokine functions such as IL-2[38] or IL-21[39] production were present.

The total magnitude of the vaccine-elicited CD4+ T cell response was also relatively low prior to challenge, although the

response was broadly directed as determined by ELISpot. It has been suggested by others that recombinant Ad5 vectors result in a suboptimal induction of CD4[+] T helper immunity during vaccination due to an absence of critical IL-2 signaling[40]. As CD4[+] T cell responses are deemed essential for control of HCV infections[11,23], insufficient priming of helper responses could have contributed to vaccine ineffectiveness. On the contrary, we observed a powerful recall and expansion (>200-fold) of RHV-specific CD4[+] T cells in the liver after viral challenge. These cells showed enhanced cytokine polyfunctionality and lost single production of TNFα. Intriguingly, this dramatic elevation in magnitude and function were not observed for virus-specific CD8[+] T cells, perhaps because substantial numbers had already migrated to the liver. Whether this change in CD4[+] T cell help is a cause or consequence of virus control remains to be determined. Nevertheless, these results indicate that conventional measures of magnitude and polyfunctionality may not necessarily predict the protective capacity of vaccine-elicited CD4[+] T cells and that the vaccine is overall effective at inducing protective helper cell immunity.

Our depletion experiments are the first to assess the relative importance of vaccine-elicited CD4[+] versus CD8[+] T cells during resolution of a primary hepacivirus infection. Our finding that CD8α-depletion prolongs RHV infection provides direct in vivo proof that this subset is critical for hepacivirus immunity. A primary antiviral role for this subset is further supported by the temporal association between recovery of blood CD8[+] T cells and re-establishment of virus control. The oscillatory pattern of viremia that preceded RHV clearance in these rats was likely a consequence of slow immune recovery paralleled by viral subversion tactics. A key question not addressed by these data is whether the protective CD8[+] T cell response expanded from an existing memory pool that survived antibody depletion or represents new responses primed by replicating antigen. Regardless of the cause, it is notable that functional CD4[+] T cell help was present in the livers of depleted animals, which likely helped facilitate antiviral CD8[+] T cell development[41]. Furthermore, the mechanisms by which CD8[+] T cells mediate control of RHV replication remains an open question. However, clearance of RHV was not associated with substantial rises in serum ALT levels, suggesting a limited role for cytolytic mechanisms. RHV infects >50% of rat hepatocytes[28], so elimination of virus via predominantly non-cytolytic mechanisms such as IFNγ[42,43] may be necessary to maintain vital liver functions.

Permanent disruption of vaccine-conferred immunity in anti-CD4 treated rats supports a central role for this subset in preventing hepacivirus persistence[11,23]. Of note, viremia was transiently suppressed in these rats for 2 weeks before failure of host immunity. This control was most likely mediated by vaccine-primed CD8[+] T cells that eventually failed in the absence of CD4[+] T cell help[41]. In the chimpanzee model, transient depletion of CD4[+] T cells led to selection of MHC class I escape variants and HCV persistence during reinfection[23]. In this study, only a single CD4-depleted rat contained a potential escape variant. This suggests that other mechanisms such as exhaustion[12,15,17,44] or deletion[45] are likely driving the failure of CD8[+] T cells in this setting. Additional studies with this model may help clarify the critical helper functions that facilitate hepacivirus clearance.

Impairment of virus-specific CD4[+] T cells is a classic feature of persistent HCV infections[5-7,11]. We demonstrated that CD4[+] T cell responses are similarly defective during experimental RHV infection. Of note, responses against the non-structural proteins, which promoted clearance in Ad-NSmut immunized animals, were undetectable or weak in unvaccinated rats infected with RHV. In contrast, delayed CD4[+] T cells specific for the structural core protein were readily detectable. These cells targeted a broad array of epitopes and produced multiple antiviral cytokines (IFNγ, TNFα, and IL-2). Interestingly, preserved responses against core are also observed during HCV persistence in humans[46-50]. Inability of these cells to contain persistent virus, their delayed induction, and their relative resistance to counter-regulatory mechanisms are not easily explained. The kinetics and ineffectiveness of these responses could be influenced by negative modulatory properties of the protein itself[51,52]. Alternatively, core may be acting as an immunological sink to attract T cell immunity away from more critical targets. Vaccination experiments involving various combinations of core and non-structural proteins could help reveal the obscure nature of this response including mechanisms that silence protective CD4[+] T cell immunity.

HCV persistence is associated with a functional silencing of virus-specific CD8[+] T cell immunity, largely attributed to a combination of mutational escape and exhaustive mechanisms[11-17]. We also found evidence of CD8[+] T cell impairment in RHV-infected rats, although of a seemingly more severe nature. CD8[+] T cells specific for RT1-A$^l$-NS3$_{1497}$ tetramer expanded between the first and second weeks of infection, a timeframe that is accelerated compared to humans where primary HCV-specific responses typically take 6–8 weeks to develop[53]. These cells expressed common markers of CD8[+] T cell activation yet showed no functionality in response to specific antigen stimulation. The response contracted sharply to undetectable levels by the third week and did not subsequently rebound during virus persistence. Mechanisms that explain the severe dysfunction and premature contraction of this virus-specific population are unclear from the present data. Notably, the NS3$_{1497}$ epitope was intact in circulating virus and tetramer-positive cells did not stain for annexin V, suggesting the influence of a non-apoptotic pathway in their disappearance. Due to the limited number of rat-specific antibodies, we were unable to assess for expression of key inhibitory markers such as PD-1, although it seems likely these negative regulatory pathways are involved. Lack of key maturation and survival signals, supplied by co-stimulation or helper cell cytokines, could stall development of the antiviral CD8+ T cell response in the liver and lead to their premature inactivation and subsequent death.

Regardless of the cause, the dysfunction and spontaneous loss of these cells represents a unique and significant observation in the context of hepacivirus infections. It has been demonstrated in the LCMV mouse model that differences in epitope affinity can lead to alternative CD8[+] T cell fates (exhaustion vs deletion) during persistent viral infection[45]. To the best of our knowledge, physical loss of an HCV-specific CD8[+] T cell response of this magnitude has not been described in chimpanzees or humans, although detection of this phenomenon could be heavily influenced by experimental constraints on epitope selection. Alternatively, the higher adaptation of RHV to rats could produce a tolerance mechanism that is more efficient in its suppression of antiviral immunity in comparison to humans. It will be necessary in future studies with this model to analyze additional class I epitopes to determine whether these CD8[+] T cell effects are a universal feature of RHV persistence or an epitope-specific event. Furthermore, a comparison of the transcriptional state of virus-specific cells after persistent RHV infection versus vaccination by next-generation sequencing technologies could reveal crucial insights into the mechanisms of CD8[+] T cell silencing and how immunization effectively reverses this outcome.

The contribution of humoral immunity to the control of acute HCV infections is unresolved. Clearance of HCV has been reported in antibody-deficient individuals[54] suggesting an ultimate dispensability. Nevertheless, broadly neutralizing anti-E1/E2 antibodies have been negatively associated with acute phase viremia[55] and develop more rapidly in those who spontaneously

resolve infection[56–58]. Selection of escape variants with mutations in neutralizing epitopes further supports a dominant role for humoral immunity in exerting negative selective pressure against replicating virus[58]. In various animal models, passive transfer of antibodies prior to experimental challenge has been shown to prevent HCV infection in some cases[59–61] while fail in others[62]. Vaccination of chimpanzees with a recombinant E1/E2 glycoprotein formulation developed by Chiron Corporation induced neutralizing antibodies and resulted in sterilizing immunity in a subset of animals[63]. Animals that developed infection after challenge showed an attenuated course of viremia and were more likely to eliminate virus, suggesting an antibody protective effect although helper T cell responses could have contributed as well. Thus, while antibodies may be unnecessary for clearance, the humoral response could nonetheless help control initial viremia and increase the likelihood of resolution mediated by other effector mechanisms.

In this regard, the RHV infection model could be invaluable in understanding the role of humoral immunity. However, assays to quantify neutralizing antibody responses to RHV glycoproteins have not yet been developed, and as such were not addressed in this study. While it is recognizable that neutralizing antibody responses against envelope proteins would not be elicited by Ad-NSmut due to the absence of these sequences in vector design, how vaccination influences subsequent humoral responses induced after infection is uncertain. Indeed, the vaccine could be operating, in part, by facilitating the development of a protective anti-E1/E2 neutralizing response that assists cellular immunity in the elimination of virus. It would be interesting in future studies to test the effectiveness of Ad-NSmut vaccination in immunoglobulin-deficient rats to better understand the potential co-operation between humoral and adaptive immunity in this protective context. Furthermore, studies of active and passive antibody vaccination in rats will be of significant value to a field devoid of a clear answer on this topic.

Despite the novelty of our findings, caution is warranted when attempting to extrapolate these data to human HCV infections. RHV and rats are sufficiently different from HCV and humans that subtle differences in infection and immunity should be expected, especially with regard to immunization strategies. It is difficult to predict whether and how differences in delivery vectors and/or adjuvants will influence the quality of a virus-specific T cell response in rodents versus humans. Indeed, the Ad5 vector utilized for this study is no longer under vaccine development for human use due to the risk of pre-existing immunity. When approaching studies with this new model, more emphasis should thus be placed on understanding what type of immunity should be elicited by vaccination to be effective rather than attempting to optimize particular delivery approaches or immunization timelines. Delineation of fundamental aspects of hepacivirus immunology, such as the role of neutralizing antibodies or helper cell requirements, will yield the most useful information necessary to build an optimal HCV vaccine for humans.

Overall, we provide compelling evidence that failure to control primary hepacivirus infections by a natural host species is the direct consequence of an inadequate T cell response. Furthermore, our data establish the functional utility of this new model for experimentally defining critical elements of hepacivirus immunity. Future studies with RHV may thus help unravel unknown mechanisms of T cell subversion and virus persistence necessary for the design of effective HCV therapies and vaccines, where strategies to sustain or revitalize CD4+ T cell responses will likely be essential for success. Furthermore, studies of RHV immunity in rats on genetically diverse backgrounds may reveal additional insights into the host factors that influence the generation and maintenance of protective immunity following natural infection or vaccination.

## Methods

**Animals.** Male and female Lewis rats were obtained from Charles Rivers Laboratories. Animals were 7–8 weeks of age at time of study initiation. All animal work received ethical approval and was monitored by the Institutional Animal Care and Use Committee of the Nationwide Children's Research Institute.

**Viral inoculum and experimental infections.** We used an RHV-rn1 infectious clone (available upon request) to amplify a large source of viral inoculum in Brown Norway rats[28]. For serum infections, rats were challenged intravenously with $10^6$ genomic equivalents (GE) of clone-derived RHV-rn1 diluted in 200 μL PBS by tail vein injection. For simplicity, RHV-rn1 is referred to as RHV throughout the text.

**Adenovirus constructs and experimental vaccinations.** A recombinant human adenovirus serotype 5 vector expressing the wild-type RHV NS3-5B coding sequence (Ad-NSmut) was generated and titrated by Vector Biolabs. Prior to cloning of vaccine insert into adenoviral vector, the GDD catalytic residue of the NS5B polymerase was mutated to GNN by site-directed mutagenesis to abolish RNA replication activity. A full list of mutagenesis and cloning primers can be found in Supplementary Table 2. For in vivo vaccinations, rats were injected once via quadriceps muscle with $5 \times 10^8$ ifu Ad-NSmut or empty control vector (Ad-null; Vector Biolabs). For testing of vaccine efficacy, rats were challenged with RHV 3 weeks after immunization.

**In vivo cell depletions.** To deplete CD8+ cells in vivo, vaccinated rats were injected intravenously with 0.5 mg anti-rat CD8α depleting (OX-8) or IgG1 isotype control antibodies (BioXcell) 2 days before virus infection. Rats received two additional doses of 0.2 mg on days 5 and 12 post infection to prolong depletion. For CD4+ cell depletions, vaccinated rats received 1.0 mg anti-rat CD4 depleting (OX-38) or IgG2a isotype control antibodies (BioXcell) on days −3, −1, and 4 of infection. Depletion and recovery of cells in blood were tracked via flow cytometry using the G28 and OX-35 antibodies which bind separate CD8 and CD4 epitopes than the OX-8 and OX-38 clones, respectively. In brief, heparinized blood was stained for 20 min with anti-rat CD3 and CD8 or CD4 antibodies, followed by direct RBC lysis and fixation (BD lyse/fix buffer).

**Quantification of viral RNA.** Serum or plasma RNA was first extracted using the QIAamp Viral RNA Mini Kit (Qiagen), followed by cDNA generation (GoScript Reverse Transcriptase, Promega) via random hexamer priming. Viral cDNA genomes were then quantified by RT-PCR using PowerUp SYBR Green Master Mix (Applied Biosystems) and the following primers: forward, 5′-TACATGGCTAA GCAATACGG-3′; reverse, 5′-AAGCGCAGCACCAATTCC-3′. The cycling conditions were as follows: 50 °C for 2 min, 95 °C for 2 min, and 40 cycles of 95 °C for 15 min, 55 °C for 15 s, and 72 °C for 1 min. A linearized molecular clone of RHV-rn1 was used as a quantitative standard.

**Serum ALT quantification.** Serum ALT levels were determined using the Max-Discovery Alanine Transaminase Enzymatic Assay Kit (Bioo Scientific).

**Leukocyte isolation, culture, and cryopreservation.** Following $CO_2$-assisted euthanasia, livers were perfused with cold PBS via inferior vena cava. For isolation of liver-infiltrating leukocytes (LILs), livers were gently homogenized through a stainless-steel mesh, followed by 37% Percoll (GE Life Sciences) gradient density centrifugation. Spleens were mechanically dissociated and filtered through a 100-μm cell strainer. For all functional assays described, cells were cultured in RPMI-1640 containing GlutaMAX and HEPES (Gibco), supplemented with 10% FBS (Gibco), 50 U/mL penicillin-streptomycin (Gibco), and 55 μM 2-mercaptoethanol (Gibco) (R10). For long-term storage, cells were cryopreserved in FBS containing 10% DMSO.

**Peptides.** One hundred and fifty-one peptides (18-aa long with 11-aa overlap) spanning the viral core, NS3, and NS4 coding regions of RHV were synthesized by Genemed Synthesis. Peptides were dissolved in sterile water containing 10% DMSO and incorporated into four testing pools (core, NS3_1, NS3_2, and NS4). The final concentration of each peptide in all functional assays was 2 μg/mL. Matrix pools for mapping responses against NS5A and NS5B were kindly provided by Charles Rice (The Rockefeller University) and tested at a concentration of 1 μg/mL per peptide. Peptides identified by matrix analysis were synthesized by Genemed Synthesis and reconstituted as described above. For testing of single peptides alone, the final concentration was 10 μg/mL. For the pool of MHC class I and II epitopes, the concentration of each peptide was 5 μg/mL. Peptides for the Ad5 hexon protein (PepTivator AdV5 Hexon) were acquired from Miltenyi Biotec and used according to manufacturer's instructions.

**IFNγ ELISpot.** Virus-specific T cells were enumerated with the anti-rat IFNγ enzyme linked immunospot (ELISpot) assay (U-Cytech) according to manufacturer's protocol. Cells were cultured at $2 \times 10^5$ cells per well in duplicate and stimulated with peptides, or media alone or Concanavalin-A (Sigma; 5 μg/mL) as

negative and positive controls, respectively, for 40–48 h prior to plate development. The total number of spot forming cells (SFCs) was calculated by subtracting the mean number of spots in the negative wells from the mean number of spots in test wells, followed by normalization to $10^6$ cells. A positive response was defined as >3 times response of background wells and >50 SFCs/$10^6$ cells after normalization.

**Quantification of intracellular cytokine production**. For detection of RHV-specific intracellular cytokine production, one million cells were stimulated in 96-well round bottom plates with peptide(s), or media alone or PMA/Ionomycin (BioLegend) as negative and positive controls, respectively, for 5-h in the presence of GolgiPlug (BD Biosciences). Following incubation, cells were surface stained for CD3, CD4, and CD8 (20 min), fixed and permeabilized using the cytofix/cytoperm kit (BD Biosciences), and intracellularly stained for IFNγ and TNFα (30 min). Dead cells were removed using the LIVE/DEAD fixable blue dead cell stain kit (Invitrogen). A positive response was defined as >3 times the background staining of the negative control sample. The percentage of cytokine positive cells was then calculated by subtracting the frequency of positive events in negative control samples from that of test samples.

**Tetramer staining**. MHC class I RT1-A$^l$ tetramers for the Lewis rat were obtained from the NIH Tetramer Core Facility. Tetramers were assessed against intrahepatic lymphocytes from Ad-NSmut immunized rats. Samples from naïve animals were used as a negative control. Cells were stained with class I tetramers for 30 min at 4 °C. After washing, cells were stained with LIVE/DEAD blue fluorescent dye and surface/intracellular markers as described above. For detection of apoptosis, cells were stained using the FITC Annexin V kit from BioLegend according to manufacturer's protocol.

**Flow cytometric analysis**. The following rat reactive antibodies (clone, catalog number, dilution) from BioLegend, BD Biosciences, and Miltenyi Biotec were used: CD3-BV605 (1F4, 563949, 1:50), CD4-AF647 (OX-35, 203312, 1:100), CD4-PerCP-eFluor710 (OX-35, 46-0040-82, 1:200), CD44H-AF647 (OX-49, 203908, 1:100), CD8-BV786 (OX-8, 740913, 1:50), granzyme B-FITC (REA226, 130-118-341, 1:10), IFNγ-FITC (DB-1, 507804, 1:50), IFNγ-AF647 (DB-1, 507810, 1:50), and TNFα-PE (TNE-19.12, 559503, 1:50). Surface and intracellular staining was performed as described above. Events were collected on a BD LSR II flow cytometer following compensation with UltraComp eBeads (Invitrogen). Data were analyzed using FlowJo v7.6.5 (Tree Star).

**Quantification of extracellular IL-2**. For analysis of IL-2 secretion, cryopreserved splenocytes were positively selected for CD4$^+$ cells using anti-rat CD4 microbeads (Miltenyi) and LS columns. Cells ($4 \times 10^6$) were plated in 1 mL of complete R10 media in single well of a 24-well plate and stimulated with 2 µg/mL peptides. After 48 h, cell culture supernatants were harvested and stored at −80 °C. Sensitive detection of IL-2 was performed using a CTLL-2 cell proliferation assay with recombinant human IL-2 (Peprotech) as a standard control. CTLL-2 cells were kindly provided by Arash Grakoui (Emory University).

**In vivo cytotoxicity assay**. To test for in vivo cytotoxicity against RHV-derived antigens, splenocytes from naïve rats were pulsed for 1 h at 37 °C with 1 µg/mL of RHV peptides (core, NS3, NS4) or the immunodominant ASYAQMTTY (ASY) epitope from Borna disease virus[64] as an irrelevant control. Following incubation, antigen-loaded cells were labeled with low (1 µM) or high (5 µM) CFSE (Invitrogen) and incubated for 15 min at 37 °C. Specific peptide-loaded cells were labeled with higher concentration of CFSE. After quenching, cells were resuspended in PBS, mixed 1:1, and 50 million were transferred intravenously via tail vein into infected rats. A non-transferred control was kept for analysis. After 18–24 h, leukocytes were harvested from livers and spleens and analyzed via FACS for CFSE expression. Percent antigen-specific killing was calculated using the following standard equation: Specific lysis = [1-(non-transferred control ratio/experimental ratio)] × 100, where ratio = frequency of CFSE$^{low}$ cells/frequency of CFSE$^{high}$ cells.

**MHC class I epitope prediction**. Potential binding motifs for the lone MHC class I RT1-A$^l$ molecule of the Lewis rat were predicted using the SYFPEITHI algorithm[31,65]. Nonamers with a binding score of >21 were selected for in vitro testing.

**Consensus PCR sequencing**. Short DNA fragments (0.5–1.2 kilobases) contained within the NS3-5B coding sequence of RHV were amplified in a single round from prepared viral cDNA using the Phusion High-Fidelity DNA Polymerase (New England Biolabs) and subsequently sequenced (Eurofins Genomics) using nested sequencing primers complimentary to the 5′ strand. A list of all primers used for amplification and sequencing reactions can be found in Supplementary Table 2.

**Statistics**. To calculate the statistical significance between vaccinated experimental groups, Fisher's exact test was employed, which analyzes the non-random associations between two categorical variables when sample sizes are small.

**Reporting summary**. Further information on experimental design is available in the Nature Research Reporting Summary linked to this article.

## Data availability
Raw data associated with Figs. 1a–g, 2a, d, 3c, d, 4c–f, and 5a–f and Supplementary Figures 3b, 4a, 5a, b, 7c, d, 8c, and d is provided as a Source Data file (https://doi.org/10.6084/m9.figshare.7707329).

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

## Acknowledgements

This study was supported by NIH grant R01AI137567 and funds from Nationwide Childrens Hospital and Research Institute. We thank Victoria M. Velazquez of Nationwide Children's Research Institute Flow Cytometry Core for technical and scientific assistance. We thank NIH Tetramer Core Facility for constructing the tetramer used in this study.

## Author contributions

A.S.H. conceived the study, performed experiments, analyzed data, and wrote the manuscript. S.M., Ar.Ku., S.T., P.D., and H.S. performed experiments and reviewed data. C.M.W. oversaw the study and provided scientific support. A.K. conceived and oversaw the study, provided scientific and financial support, and wrote the manuscript.

## Additional information

**Competing interests:** The authors declare no competing interests.

