## [Peer Review File · Nature Communications]

Reviewers' Comments:

Reviewer #1:

Remarks to the Author:

In this review article entitled "Vaccination to prevent T cell subversion can protect against persistent hepacivirus infection" Alex Hartlage et al. elegantly use a small animal model for hepatitis C virus (HCV) infection. Indeed, previously they described an HCV rodent hepacivirus (RHV) that is able to infect laboratory mice and rats. This virus persists in rats despite an activation on innate and adaptive immunity. In this study, the authors have used this model to better determine the characteristics of vaccine induced protective immunity. They could show that vaccination with an adenovirus expressing non-structural virus proteins can prevent the occurrence of viral persistence in most animals.

Furthermore, by performing depletion studies of CD4+ versus CD8+ T cell subsets they could clearly show that CD8+ T cells are primarily responsible for viral control, however, that in the absence of CD4+ T cell help viral persistence occurs. These findings are similar to observations made in the HCV chimpanzee model, partly by the same group – however, displaying some important differences:

1. After challenge with RHV, virus-specific CD8+ T cells are rapidly deleted and do not induce viral escape.
2. Virus-specific CD4+ T cell responses are detectable, however, primarily targeting the core region.
3. After CD4+ depletion, virus-specific CD8+ T cell failure and viral persistence are not associated with the emergence of viral escape mutations.

Overall, by using a novel small animal model this study clearly shows that vaccination is clearly able to prevent RHV persistence and that this is due to the induction of a sufficient virus-specific T-cell response. The study is very well performed and state-of-the-art immunological assays, such as tetramers have been used for the visualization of the virus-specific T cell response. The data is well displayed and discussed.

Comments:

1. The tetramer data shown in figure 1 E. is convincing; it would be good, however, to perform a more thorough phenotypical and transcriptional characterization of these cells, especially, as they seem to disappear in the course of infection. Are they highly exhausted, activated or show markers of apoptosis? Also, a similar analysis of the CD8+ T cell responses after CD4 depletion should be performed to gain better insights into the mechanism of CD8+ T cell failure after CD4 depletion.
2. In this reviewer's opinion, it would be helpful to add a table showing the results obtained from the RHV model versus the HCV chimpanzee model to highlight similarities and obvious differences in the discussion.
3. Fig.1I, in this reviewer's opinion, data from more animals should be displayed in this figure.

Reviewer #2:

Remarks to the Author:

The manuscript by Hartlage et al. describes a first description of natural and vaccine-induced T cell immunity in the novel rat hepacivirus model. The authors conclude that both CD4 and CD8 T cells are critical for the control of this mostly chronic infection, and that such responses can be readily induced by a adenovirus-based rat hepacivirus vaccine.

Overall, this is a very important and well designed study. Strength are the novel and truly unique hepacivirus mouse model, allowing to analyze immunity in a natural viral infection in its natural host. The latter is critical and a key advantage of this model over the same infection in mice, as infection usually leads to chronicity without manipulation of the immune system, and because the model reflects long adaption of the virus to its natural host. The experiments are well designed and allow to

fully evaluate the results. My comments and questions mostly relate to discussion and interpretation of the data, specifically:

1) line 100 states that nearly 100% of the rats developed chronicity. What is the exact number? And what would it take to test significant numbers of rats with resolving infection? This would not only be a very nice control group for both the chronic and the vaccinated animals. More importantly this would be to my knowledge the very first rodent model of viral infection that can analyze both chronic and acute infection without any manipulation of virus or host (as is required in LCMV and other classic mouse models). In my opinion, such comparison would be even more exciting than the vaccine studies. These opportunities could be discussed.

2) There should be some more discussion about the relevance of further refined vaccination studies in this model. While the natural infection model is really outstanding, because of the infection taken place in the host to which the virus has adapted to, the vaccination studies might be only partially translatable into humans. It is really hard to predict whether and how certain vectors, adjuvants or even just vaccination timelines translate into humans. As said above, I think this model is outstanding for understanding basic immunology and also for determining what kind of immunity a vaccine needs to induce to be protective. I would be more cautious about using it to decide what kind of vaccination approach is best suited to induce such immunity in humans, and these distinctions could be discussed more.

3) There could be more discussion about the fact that almost no functional CD8 responses were present in the liver, but robust tetramer responses were detected. Is this because at the site of infection all the cells are already highly stimulated *in vivo*? Would the results be different if T cells were studied in the blood? Clearly, in human HCV infection, rather significant IFN-secreting CD8 and CD4 populations targeting mostly NS proteins can be detected in the blood, even in patients developing chronicity (if tested reasonably early in infection).

And how do the authors know that the tetramer epitope is immunodominant (line 119), given that their Elispot screening was all negative. Did they screen the immune response by other means, such as short term cell lines? Or was this information just deferred from the vaccination results?

4) The vaccination results need some clarification. Clearly based on extended figure 1, CD4 responses were much more broad and vigorous than CD8 responses, at least in the spleen. This is different from the liver, where more CD8 responses were detected. But is this difference between tissues not to be expected, given the different roles of these cells? That overall the vaccine is very good at inducing CD4 responses is shown during the challenge.

The latter also demonstrates why one should be cautious about directly using the data for refining human vaccine studies (apart from determining what kind of immunity a vaccine should induce), as it seems that the Ad vector is much better at inducing CD4 responses in rats vs chimps and humans. I also think the statement on line 144 should be qualified, based on the good CD4 responses after vaccination in the spleen.

5) Some minor issues:

line 211: it seems to me that both CD4 and CD8 responses are equally critical

line 217: is T cell vaccination really unique for HCV? there have been multiple studies in HIV

line 253: again, I am not in full agreement with the statement that the CD4 response was much weaker. After vaccination and pre infection it seems to be expected that responses do not migrate to the liver (why should they), but rather circulate in lymphatic organs. From the spleen data, the CD4 response seems dominant.

line 60ff: the classic chimps experiments with CD4 and CD8 depletion by Dr. Walker are not really mentioned in the discussion of evidence for the role of T cell responses in protection from chronic HCV

infection. These have established the critical role for T cells, contradicting the statement in line 70 lines 220ff: I do not think the current experiment established the efficacy of vaccination any more than the published chimp data. Both experiments showed significantly better control after vaccination, and both also showed that even with an identical or very similar challenge not all animals were protected. The latter should also be more discussed in detail, given that one really has to wonder about the efficacy against heterologous challenges.

line 305: it is not rare that CD8 responses disappear from the blood in chronic HCV infection after the acute phase (though often they can still be detected after magnetic bead tetramer enrichment). Since the assay described in this manuscript was not designed to determine complete deletion of the population, this comment should be deleted.

Reviewer #3:

Remarks to the Author:

This is a very nice study of vaccination against a rat hepatitis virus using a T cell vaccine. This virus is a distant relative of the human HCV and causes persistent infections in the rat as does HCV in humans and so this is a very interesting animal model for HCV for which there is an urgent need for a vaccine and the absence of a suitable animal model.

The authors show that vaccination with a defective adenovirus expressing the viral nonstructural proteins is able to prime CD4 & CD8+ T cells against the virus and that these inhibit viral load following challenge and lead to viral eradication unlike in unvaccinated control rats. They also show that removal of either CD4 or CD8+ T cells by antibody depletion abrogates the protective effects of the vaccine. However, in the case of CD4 cells, their re-emergence following antibody decline is not able to control viremia unlike the situation with CD8 cells. Furthermore, the authors show that in unvaccinated animals, viral persistence occurs despite strong CD4 T cell responses to the viral core protein.

I believe that this ms is of great value to the field of HCV and vaccinology in general but that it would be of further value to these fields if it addresses the following points :

1. Despite 6/9 vaccinated animals resolving viremia following challenge as compared with 0/6 unvaccinated controls, the authors imply that this is not statistically significant. Proving that such a T cell vaccine can really be protective against chronic infections is of the utmost importance and so the ms would greatly benefit from showing statistical significance in this regard. Furthermore, the authors should point out that when they refer to the related human HCV T cell vaccine work in ref 25, that there is no evidence that this vaccine reduced chronicity in the chimpanzee model (it did not despite ameliorating acute infection and acute hepatitis).
2. The T cell vaccine could be working in part by helping anti-envelope neutralizing antibody responses but this important topic is not addressed at all and neither is the acquired evidence that human HCV infection is controlled in part by humoral immunity. These aspects need to be brought out in a revised ms.

Reviewers' comments:

Reviewer #1 (Remarks to the Author):

In this review article entitled “Vaccination to prevent T cell subversion can protect against persistent hepatitis C virus infection” Alex Hartlage et al. elegantly use a small animal model for hepatitis C virus (HCV) infection. Indeed, previously they described an HCV rodent hepatitis virus (RHV) that is able to infect laboratory mice and rats. This virus persists in rats despite an activation on innate and adaptive immunity. In this study, the authors have used this model to better determine the characteristics of vaccine induced protective immunity. They could show that vaccination with an adenovirus expressing non-structural virus proteins can prevent the occurrence of viral persistence in most animals. Furthermore, by performing depletion studies of CD4+ versus CD8+ T cell subsets they could clearly show that CD8+ T cells are primarily responsible for viral control, however, that in the absence of CD4+ T cell help viral persistence occurs. These findings are similar to observations made in the HCV chimpanzee model, partly by the same group – however, displaying some important differences:

1. After challenge with RHV, virus-specific CD8+ T cells are rapidly deleted and do not induce viral escape.
2. Virus-specific CD4+ T cell responses are detectable, however, primarily targeting the core region.
3. After CD4+ depletion, virus-specific CD8+ T cell failure and viral persistence are not associated with the emergence of viral escape mutations.

Overall, by using a novel small animal model this study clearly shows that vaccination is clearly able to prevent RHV persistence and that this is due to the induction of a sufficient virus-specific T-cell response. The study is very well performed and state-of-the-art immunological assays, such as tetramers have been used for the visualization of the virus-specific T cell response. The data is well displayed and discussed.

Comments:

1. The tetramer data shown in figure 1 E. is convincing; it would be good, however, to perform a more thorough phenotypical and transcriptional characterization of these cells, especially, as they seem to disappear in the course of infection. Are they highly exhausted, activated or show markers of apoptosis? Also, a similar analysis of the CD8+ T cell responses after CD4 depletion should be performed to gain better insights into the mechanism of CD8+ T cell failure after CD4 depletion.

We agree fully with the reviewer's suggestions. We have updated the manuscript to include an additional characterization of the tetramer response in infected animals, assessing for markers of activation and apoptosis. While a transcriptional profile of CD8+ T cells in vaccinated versus unvaccinated animals would be highly informative, we feel that this type of experiment and analysis is beyond the scope of the manuscript at this present time, especially since we have only a single working tetramer currently and next generation sequencing approaches will need to be employed in the absence of extensive flow antibodies or microarrays for rat. The main demonstration of the tetramer data is that non-functioning CD8+ T cells are primed by replicating virus but rapidly fail during virus persistence, making RHV a relevant HCV model for testing T cell vaccination strategies.

2. In this reviewer's opinion, it would be helpful to add a table showing the results obtained from the RHV model versus the HCV chimpanzee model to highlight similarities and obvious differences in the discussion.

While we agree a graphical comparison would be useful, this type of data feels better suited for a review or expert commentary, especially since the RHV model is still in its early phases. The most

obvious difference that has been elicited at this time is in the nature of the CD8+ T cell response which we feel has been sufficiently highlighted and discussed in the manuscript.

3. Fig.11, in this reviewer's opinion, data from more animals should be displayed in this figure.

The manuscript has been updated to include additional animals and timepoints for a more thorough analysis.

Reviewer #2 (Remarks to the Author):

The manuscript by Hartlage et al. describes a first description of natural and vaccine-induced T cell immunity in the novel rat hepacivirus model. The authors conclude that both CD4 and CD8 T cells are critical for the control of this mostly chronic infection, and that such responses can be readily induced by an adenovirus-based rat hepacivirus vaccine.

Overall, this is a very important and well-designed study. Strength are the novel and truly unique hepacivirus mouse model, allowing to analyze immunity in a natural viral infection in its natural host. The latter is critical and a key advantage of this model over the same infection in mice, as infection usually leads to chronicity without manipulation of the immune system, and because the model reflects long adaption of the virus to its natural host. The experiments are well designed and allow to fully evaluate the results. My comments and questions mostly relate to discussion and interpretation of the data, specifically:

1) line 100 states that nearly 100% of the rats developed chronicity. What is the exact number? And what would it take to test significant numbers of rats with resolving infection? This would not only be a very nice control group for both the chronic and the vaccinated animals. More importantly this would be to my knowledge the very first rodent model of viral infection that can analyze both chronic and acute infection without any manipulation of virus or host (as is required in LCMV and other classic mouse models). In my opinion, such comparison would be even more exciting than the vaccine studies. These opportunities could be discussed.

As described in our first manuscript on RHV, spontaneous virus clearance has only been observed in a single rat from an outbred stock (Holtzman). All other rat strains, including Lewis, have developed chronic infection at a 100% incidence rate. Focusing on a single strain with a shared haplotype and total susceptibility to chronic infection made for a cleaner analysis of immunity and protective vaccination, although the reviewer's points are well received. A more thorough discussion here of the ideas raised by the reviewer might distract reviewers away from the main message of the paper and can already be found in our first article.

2) There should be some more discussion about the relevance of further refined vaccination studies in this model. While the natural infection model is really outstanding, because of the infection taken place in the host to which the virus has adapted to, the vaccination studies might be only partially translatable into humans. It is really hard to predict whether and how certain vectors, adjuvants or even just vaccination timelines translate into humans. As said above, I think this model is outstanding for understanding basic immunology and also for determining what kind of immunity a vaccine needs to induce to be protective. I would be more cautious about using it to decide what kind of vaccination approach is best suited to induce such immunity in humans, and these distinctions could be discussed more.

We agree translatability of vaccine approach is a major uncertainty of the model. Our major focus, as the reviewer correctly highlights, is in understanding what kind of immunity a vaccine needs to induce to be effective. We have updated the discussion section to further highlight this focus of the model.

3) There could be more discussion about the fact that almost no functional CD8 responses were present in the liver, but robust tetramer responses were detected. Is this because at the site of infection all the cells are already highly stimulated in vivo? Would the results be different if T cells were studied in the blood? Clearly, in human HCV infection, rather significant IFN-secreting CD8 and CD4 populations targeting mostly NS proteins can be detected in the blood, even in patients developing chronicity (if tested reasonably early in infection).

We could not identify functional responses in spleen either despite tetramer positivity, so site of infection and antigen expression does not seem to affect whether a functional response is detected. As further described in the discussion, we believe CD8+ T cells exhibit a greater functional impairment in this model compared to HCV and likely completely fail to functionally mature.

And how do the authors know that the tetramer epitope is immunodominant (line 119), given that their Elispot screening was all negative. Did they screen the immune response by other means, such as short term cell lines? Or was this information just deferred from the vaccination results?

CD4 and CD8 epitope identification was identified using vaccinated animals. While there is potential for a difference between infected vs vaccinated rats in terms of epitope preference and hierarchy, this risk is overall minimal and not limiting since a functional tetramer was ultimately made. We are in the process of constructing additional tetramers to more comprehensively analyze the CD8 response.

4) The vaccination results need some clarification. Clearly based on extended figure 1, CD4 responses were much more broad and vigorous than CD8 responses, at least in the spleen. This is different from the liver, where more CD8 responses were detected. But is this difference between tissues not to be expected, given the different roles of these cells? That overall the vaccine is very good at inducing CD4 responses is shown during the challenge.

In our technical experience, CD4 ELISpot responses produce very large spots in comparison to CD8 responses, which is why it is a great assay for mapping yet can give the misleading impression that they are more vigorous in this experimental setting (and why it is a supplemental figure). When analyzed on a per cell basis by flow, it is quite clear that the magnitude of the IFN γ + CD8 response is much greater in the liver and fairly comparable in spleen. Moreover, if you compare the CD4 responses between liver and spleen in Figure 4, it is clear that there is no magnitude difference for CD4s after vaccination. We agree that the CD4 response is broader, but that tends to be a feature of helper responses in general. Nevertheless, we have updated the manuscript to de-emphasize this difference, since as the reviewer rightly argues, the localization of CD8s to the liver is probably expected based on the role of these cells. We do however believe the magnitude change in CD4 helper responses after challenge is an important point to emphasize as it suggests an active, critical role for this subset and justifies the deletion experiments.

The latter also demonstrates why one should be cautious about directly using the data for refining human vaccine studies (apart from determining what kind of immunity a vaccine should induce), as it seems that the Ad vector is much better at inducing CD4 responses in rats vs chimps and humans. I also think the statement on line 144 should be qualified, based on the good CD4 responses after vaccination in the spleen.

We concur with the reviewer. The line in question has been modified to simply report the magnitude of responses in liver for both subsets without overly qualifying them.

5) Some minor issues:

line 211: it seems to me that both CD4 and CD8 responses are equally critical

We have updated the manuscript to refine our initial statement.

line 217: is T cell vaccination really unique for HCV? there have been multiple studies in HIV.

While we agree with the reviewer that T cell vaccines have been tested previously for HIV and HCV, their usage is still unconventional from the perspective of the greater vaccine field. We feel this is an important point to emphasize since virtually all vaccines in use today induce neutralizing antibody responses and here we have completely eliminated their influence from vaccine design.

line253: again, I am not in full agreement with the statement that the CD4 response was much weaker. After vaccination and pre infection it seems to be expected that responses do not migrate to the liver (why should they), but rather circulate in lymphatic organs. From the spleen data, the CD4 response seems dominant.

We have updated the manuscript to reflect this concern.

line 60ff: the classic chimps experiments with CD4 and CD8 depletion by Dr. Walker are not really mentioned in the discussion of evidence for the role of T cell responses in protection from chronic HCV infection. These have established the critical role for T cells, contradicting the statement in line 70

We acknowledge the reviewer's concern here. The discussion of previous experiments was intended to emphasize the critical role for T cells during the primary phase infection. The depletion experiments performed by Dr. Walker were in immune chimpanzees that had already controlled the virus previously and were responding to a secondary challenge. This is a subtle difference that we believe is worth highlighting. We have updated the manuscript to highlight and communicate this difference more clearly.

lines220ff: I do not think the current experiment established the efficacy of vaccination any more than the published chimp data. Both experiments showed significantly better control after vaccination, and both also showed that even with a identical or very similar challenge not all animals were protected. The latter should also be more discussed in detail, given that one really has to wonder about the efficacy against heterologous challenges.

While we acknowledge the reviewer's perspective, we agree with reviewer 3 on the interpretation of the chimpanzee vaccination data. The chimp data established that T cell vaccination could: 1) reduce peak viral titers, and 2) reduce infection duration (accelerated clearance). Whether vaccination could reduce the rate of viral chronicity could not conclusively be assessed because of too few animals. Furthermore, chimpanzees are inherently prone to spontaneously clear HCV, so interpreting vaccination data is always troublesome. Our data, which has now been updated with statistics, is the first to draw a statistical conclusion regarding the efficacy of a T cell vaccine in a natural host model, where the goal is to significantly reduce hepatitis persistence.

line 305: it is not rare that CD8 responses disappear from the blood in chronic HCV infection after the acute phase (though often they can still be detected after magnetic bead tetramer enrichment). Since the assay described in this manuscript was not designed to determine complete deletion of the population, this comment should be deleted.

We have updated the manuscript to highlight that the responses are no longer detectable by direct tetramer staining and simply refer to the event as a premature contraction without assuming complete deletion since it was not specifically tested for in this setting as the reviewer rightly points out.

Reviewer #3 (Remarks to the Author):

This is a very nice study of vaccination against a rat hepacivirus using a T cell vaccine. This virus is a distant relative of the human HCV and causes persistent infections in the rat as does HCV in humans and so this is a very interesting animal model for HCV for which there is an urgent need for a vaccine and the absence of a suitable animal model.

The authors show that vaccination with a defective adenovirus expressing the viral nonstructural proteins is able to prime CD4 & CD8+ T cells against the virus and that these inhibit viral load following challenge and lead to viral eradication unlike in unvaccinated control rats. They also show that removal of either CD4 or CD8+ T cells by antibody depletion abrogates the protective effects of the vaccine. However, in the case of CD4 cells, their re-emergence following antibody decline is not able to control viremia unlike the situation with CD8 cells. Furthermore, the authors show that in unvaccinated animals, viral persistence occurs despite strong CD4 T cell responses to the viral core protein.

I believe that this ms is of great value to the field of HCV and vaccinology in general but that it would be of further value to these fields if it addresses the following points:

1. Despite 6/9 vaccinated animals resolving viremia following challenge as compared with 0/6 unvaccinated controls, the authors imply that this is not statistically significant. Proving that such a T cell vaccine can really be protective against chronic infections is of the utmost importance and so the ms would greatly benefit from showing statistical significance in this regard. Furthermore, the authors should point out that when they refer to the related human HCV T cell vaccine work in ref 25, that there is no evidence that this vaccine reduced chronicity in the chimpanzee model (it did not despite ameliorating acute infection and acute hepatitis).

We have updated the manuscript to include a test for statistical difference. Furthermore, we have emphasized the limitations of the chimpanzee data which, as the reviewer rightly comments, does not achieve the main objective of a T cell vaccine against HCV.

2. The T cell vaccine could be working in part by helping anti-envelope neutralizing antibody responses but this important topic is not addressed at all and neither is the acquired evidence that human HCV infection is controlled in part by humoral immunity. These aspects need to be brought out in a revised ms.

We have updated the manuscript to include a discussion on antibody responses in HCV and this model.

Reviewers' Comments:

Reviewer #1:

Remarks to the Author:

The authors have carefully addressed my concerns, e.g. by showing additional data. I also agree with their comment about the table comparing their data to the HCV chimpanzee studies. I have no further comments.

Reviewer #2:

Remarks to the Author:

The authors have provided convincing answers to the points raised in the original reviews. As said before, is is an important study for the field.